# Relationship between the Blood Urea Nitrogen to Creatinine Ratio and In-Hospital Mortality in Non-Traumatic Subarachnoid Hemorrhage Patients: Based on Propensity Score Matching Method

**DOI:** 10.3390/jcm11237031

**Published:** 2022-11-28

**Authors:** Zirong Chen, Junhong Wang, Hongkuan Yang, Hua Li, Rudong Chen, Jiasheng Yu

**Affiliations:** Department of Neurosurgery, Tongji Hospital, Tongji Medical College, Huazhong University of Science and Technology, Wuhan 430030, China

**Keywords:** non-traumatic subarachnoid hemorrhage, blood urea nitrogen to creatinine ratio, in-hospital mortality, MIMIC-Ⅳ database

## Abstract

(1) Background: To explore the correlation between the blood urea nitrogen to creatinine ratio (UCR) and in-hospital mortality in non-traumatic subarachnoid hemorrhage patients. (2) Methods: Specific clinical information was collected from the Medical Information Mart for Intensive Ⅳ (MIMIC-Ⅳ) database. The optimal cut-off value of the UCR was calculated with ROC curve analysis conducted using the maximum Youden index for the prediction of survival status. Univariable and multivariable logistic regression analyses were also carried out to assess the prognostic significance of UCR, and the Kaplan–Meier (K–M) analysis was conducted to draw the survival curves. Then, the 1:1 propensity score matching (PSM) method was applied to improve the reliability of the research results while balancing the unintended influence of underlying confounders. (3) Results: This retrospective cohort study included 961 patients. The optimal cut-off value of the UCR for in-hospital mortality was 27.208. The PSM was performed to identify 92 pairs of score-matched patients, with balanced differences exhibited for nearly all variables. According to the K–M analysis, those patients with a UCR of more than 27.208 showed a significantly higher level of in-hospital mortality compared to the patients with a UCR of less than 27.208 (*p* < 0.05). After the adjustment for possible confounders, those patients whose UCR was more than 27.208 still had a significantly higher level of in-hospital mortality than the patients whose UCR was less than 27.208, as revealed by the multivariable logistic regression analysis (OR = 3.783, 95% CI: 1.959~7.305, *p* < 0.001). Similarly, the in-hospital mortality remained substantially higher for those patients in the higher UCR group than for the patients in the lower UCR group after PSM. (4) Conclusion: A higher level of the UCR was evidently associated with an increased risk of in-hospital mortality, which made the ratio useful as a prognostic predictor of clinical outcomes for those patients with non-traumatic subarachnoid hemorrhage.

## 1. Introduction

Non-traumatic subarachnoid hemorrhage is a neurological emergency mainly caused by the rupture of intracranial aneurysms. It requires timely diagnosis and effective management to prevent life-threatening rebleeding and improve the prognosis. The incidence of rebleeding can be effectively controlled by surgical clipping or endovascular therapy. Although the mortality rate for aneurysmal subarachnoid hemorrhage has decreased significantly in the past decade, the rate remains higher than 30% [1]. Given the risk of subarachnoid hemorrhage, it is necessary to find non-invasive and inexpensive tests to identify those at greater risk of death and to further reduce the mortality rate.

Blood urea nitrogen (BUN) is produced by the liver and excreted by the kidneys. It is a biomarker that can reflect the function of the liver and kidneys. Recently, it was reported that the BUN level is an effective prognostic factor for many cases, including ischemic stroke [2], chronic obstructive pulmonary disease [3], cardiogenic shock [4], acute ST-elevation myocardial infarction [5], neonatal sepsis [6], and bone marrow transplants [7]. The level of creatinine is commonly used to reflect the renal function, which can help to judge whether the renal function is in a stage of potential failure or improvement [8]. However, the level of BUN and creatinine can be affected by many factors, such as the use of corticosteroids, protein intake, and dehydration. Therefore, the BUN/creatinine ratio (UCR) is a relatively useful parameter, which can reduce the effect of the above factors. In recent studies, it was reported that an elevated UCR is a poor prognosis factor for patients with septic shock [8], ischemic stroke [2,9,10], acute heart failure [11], and chronic heart failure [12].

To our knowledge, this is the first study to assess the relationship between the UCR and in-hospital mortality of subarachnoid hemorrhage patients. Therefore, the present study is aimed at exploring the prognostic significance of the UCR in subarachnoid hemorrhage patients and providing a simple and convenient indicator for high-risk patients.

## 2. Materials and Methods

### 2.1. Data Sources

We extracted data from the Medical Information Mart for Intensive Care (MIMIC)-IV [13], a free and publicly available database. We were allowed to extract data after we completed the training courses regulated by the National Institutes of Health (NIH) and the Protecting Human Research Participants examination. One author, Junhong Wang, was approved to utilize the database. Our study was also approved by the Institutional Review Boards of Beth Israel Deaconess Medical Center and the Massachusetts Institute of Technology (Cambridge, MA, USA). Additional ethical approval was not needed. The findings of this study are reported following the Strengthening the Reporting of Observational Studies in Epidemiology guidelines [14].

### 2.2. Study Population

A total of 1277 patients with non-traumatic subarachnoid hemorrhage were selected. The diagnosis of subarachnoid hemorrhage was based on the International Classification of Disease, Ninth and Tenth Revision. We selected patients who met the following standards: (1) those who were first admitted to the ICU; (2) those whose age was over 18 years; and (3) those who finished a UCR examination in the first 24 h of being admitted to the ICU. ICU patients with length of stay was less than 24 h also excluded to avoid potential extremum value influence. We excluded patients according the following standard: the length of ICU stay was less than 24 h. Thus, only 961 patients were included in this study. The workflow is shown in Figure 1.

### 2.3. Data Extraction

We extracted variables from the MIMIC-IV database: (1) demographics: sex, age, and ethnicity; (2) vital signs: systolic blood pressure (SBP), respiratory rate (RR), heart rate (HR), diastolic blood pressure (DBP), temperature, and percutaneous oxygen saturation (SpO2); (3) comorbidities: myocardial infarction, congestive heart failure (CHF), chronic pulmonary disease (CPD), delayed cerebral ischemia (DCI), peripheral vascular disease, mild liver disease, diabetes, etc.; (4) laboratory results: white blood cell count (WBC), international normalized ratio (INR), neutrophil count, monocyte count, activated partial thromboplastin time (APTT), prothrombin time (PT), and aspartate transaminase (AST). In addition, the first laboratory test results for blood urea nitrogen and creatinine values after ICU admission were extracted as the interest variable and the major exposure factor in this study. The sequential organ failure assessment (SOFA) score, Oxford Acute Severity of Illness Score (OASIS), World Federation of Neurosurgical Societies (WFNS), and Glasgow Coma Scale (GCS) were considered to measure the admission severity.

### 2.4. Endpoints

The primary outcome was in-hospital mortality. The secondary outcomes were the ICU stay length and hospital stay length.

### 2.5. Statistical Analysis

The continuous variables were displayed as the average ± standard deviation (SD) or the mid-value (interquartile range). The Student’s *t*-test or Mann–Whitney U-test was used according to the normality of the data distribution. Categorical variables were displayed as a case quantity (%), and the chi-square test (or Fisher’s exact approach) was utilized for analyses.

The optimal cut-off value of the UCR was calculated with ROC curve analysis conducted using the maximum Youden index for the prediction of survival status. The Youden index = sensitivity + specificity − 1. The UCR was divided into two groups based on the cut-off value.

Univariable and multivariable regression analyses were carried out to assess the prognostic significance of the UCR. The screening criteria for confounders included: (1) a factor affected the research variable (with impact over 10%); (2) the outcome variables might be obviously impacted by some factors based on previous experiences; and (3) the univariable analysis revised the variables, with *p* less than 0.05. 

The crude model did not adjust any of the variables. In the multivariable analysis, we performed different statistical models to verify the stability of the results. Model I made adjustments to the variables of age, gender, and ethnicity. Model Ⅱ made adjustments to 6 variables, including myocardial infarction, congestive heart failure, renal disease, mild liver disease, diabetes, and sepsis. Model Ⅲ made further adjustments to 15 variables, including HR, RR, platelets, WBC, anion-gap, bicarbonate, chloride, sodium, INR, PT, APTT, OASIS, GCS, WFNS, and SOFA.

Given the difficulty of achieving complete stochasticity for the screening of patients, the PSM approach was used to balance the influence of selection bias and underlying confounders. The PSM analysis was conducted with the logistic regression model developed using age, sex, ethnicity, HR, DBP, MBP, temperature, etc. A standardized mean difference (SMD) was used to examine the PSM degree, and a lower threshold than 0.1 was treated as acceptable. For the pairs of patients with a low UCR (<27.208) and a high UCR (≥27.208), 1:1 matching was performed with a caliper of 0.1. Finally, 184 propensity score-matched patients and 92 pairs of score-matched patients were identified.

The estimated propensity scores were used as weights. Pairwise algorithmic (PA) [15], standardized mortality ratio weight (SMRW) [16], inverse probability of treatment weight(IPTW), [17] and overlap weight (OW) [18] were used to generate a weighted cohort to adjust the baseline confounders. The weighted cohort could accurately reflect the independent association between the UCR and in-hospital mortality. 

The subgroup analysis was conducted to determine how the UCR affected the in-hospital mortality from various perspectives including age (<65 and ≥65 years old), sex, myocardial infarction, congestive heart failure, peripheral vascular disease, chronic pulmonary disease, renal disease, malignant cancer, mild liver disease, diabetes, SOFA (<3 and ≥3), sepsis, and WFNS grade. We conducted the subgroup analyses using a logistic regression model.

The statistic program packages R 3.3.2 (http://www.R-project.org, The R Foundation) and Free Statistics software version 1.4 (Beijing, China) were used to complete all analyses. The study carried out a two-tailed test and *p* < 0.05 was statistical significance. 

## 3. Results

### 3.1. Data Sources

We selected patients who met the preset standards (see Figure 1 for a flow chart). 

### 3.2. Clinical Characteristics of Study Subjects

Table 1 compares the demographic data, vital signs, comorbidities, treatment, laboratory results, scores, and outcomes between the survivor and non-survivor patient groups. Overall, the median age of patients was 60.0 years old, and approximately 56.0% were women. The non-survivor group presented a higher UCR than the survivor group (median: 18.0 vs. 16.7, respectively, *p* = 0.006). Compared to the survivor group, the non-survivor group was older (68.0 vs. 58.5 years old, respectively, *p* < 0.001), and presented a higher comorbidity incidence of myocardial infarction, congestive heart failure, chronic pulmonary disease, renal disease, mild liver disease, diabetes, and sepsis as well as higher OASIS and lower GCS scores (all *p* values < 0.05). The levels of urea nitrogen, creatinine, INR, PT, APTT, glucose, WBC, and anion gap in non-survivor group were significantly higher than survivor group (Table 1).

### 3.3. The Prognostic Significance of UCR before PSM

The ROC curve of the UCR was plotted, and the AUC was 0.564 (95% CI, 0.515–0.613) (Appendix A). The best cut-off value of the UCR was calculated with ROC curve analysis, using the highest Youden index to predict the survival status, where the Youden index = sensitivity + specificity − 1. The corresponding optimal cut-off value was 27.208, the evaluation sensitivity was 21.7%, and the specificity was 29.0% (Appendix A). Based on the cut-off value, 961 patients were divided into the low UCR (<27.208, n = 865) group and the high UCR (≥27.208, n = 96) group. The demographics, coexisting diseases, vital signs, scoring, laboratory results, etc. are presented in Table 2. Compared to patients in the low UCR (<27.208) group, patients in the high UCR (≥27.208) group were at higher risk of in-hospital mortality (42.7 vs. 17.1%, respectively, *p* < 0.001) and had a higher comorbidity incidence for myocardial infarction, congestive heart failure, dementia, renal disease, malignant cancer, and diabetes (*p* < 0.05) (Table 2).

### 3.4. Association between UCR and in-Hospital Mortality in Non-Traumatic Subarachnoid Hemorrhage Patients before PSM

Appendix A and Table 3 list the univariable and multivariable logistic analysis results separately. Table 3 shows an unadjusted and a multivariable-adjusted correlation between the UCR and in-hospital mortality. 

Before PSM, as a continuous variable, the UCR was positively related to the in-hospital mortality (Crude Model: OR = 1.038, 95% CI: 1.018–1.059, *p* < 0.001; Model I: OR = 1.030, 95% CI: 1.008–1.052, *p* = 0.0071; Model Ⅱ: OR = 1.031, 95% CI: 1.009–1.054, *p* = 0.0062; Model Ⅲ: OR = 1.038, 95% CI: 1.009–1.068, *p* = 0.0102). Moreover, as a categorical variable, the in-hospital mortality increased remarkably for patients in the high UCR (≥27.208) group compared to the low UCR (<27.208) group (Crude Model: OR = 3.611, 95% CI: 2.323–5.615, *p* < 0.001; Model Ⅰ: OR = 3.110, 95% CI: 1.937–4.995, *p* < 0.001; Model Ⅱ: OR = 2.979, 95% CI: 1.818–4.844, *p* < 0.001; Model Ⅲ: OR = 3783, 95% CI: 1.959–7.305, *p* < 0.001) (Table 3).

Figure 2 displays the K–M curves of the two groups. The high UCR (≥27.208) group exhibited remarkably higher in-hospital mortality (Figure 2A) compared to the low UCR (<27.208) group (*p* < 0.001).

### 3.5. The Results of PSM

Considering that the two groups presented imbalanced baseline features, a 1:1 ratio PSM was completed to balance the latent confounders, which obtained 92 pairs of score-matched sufferers. The difference between the two groups were balanced in terms of nearly all variables, and a favorable matching performance was achieved (Figure 3).

### 3.6. The Clinical Characteristics of Non-Traumatic Subarachnoid Hemorrhage Patients after PSM

The clinical characteristics of non-traumatic subarachnoid hemorrhage patients after PSM are shown in Table 4. After PSM, the high UCR group (≥27.208) still presented obvious higher in-hospital mortality than the low UCR group (<27.208) (43.5 vs. 20.7%, respectively, *p* < 0.001) (Table 4).

### 3.7. Association between UCR and in-Hospital Mortality in Non-Traumatic Subarachnoid Hemorrhage Patients after PSM

After PSM, as a continuous variable, the UCR was still positively related to the in-hospital mortality (Crude Model: OR = 1.039, 95% CI: 1.008–1.071, *p* = 0.0014; Model I: OR = 1.036, 95% CI: 1.003–1.070, *p* = 0.0326; Model Ⅱ: OR = 1.041, 95% CI: 1.006–1.078, *p* = 0.0215; Model Ⅲ: OR = 1.0661, 95% CI: 1.011–1.124, *p* = 0.0189) (Table 3).

As a categorical variable, the in-hospital mortality still increased remarkably for patients in the high UCR (≥27.208) group compared to the low UCR (<27.208) group (Crude model: OR= 2.995, 95% CI: 1.540–5.671, *p* = 0.011; Model Ⅰ: OR = 3.082, 95% CI: 1.515–6.271, *p* = 0.019; Model Ⅱ: OR = 3.634, 95% CI: 1.673–7.892, *p* = 0.0011; Model Ⅲ: OR = 10.161, 95% CI: 2.691–38.368, *p* < 0.001) (Table 3).

Figure 2 displays the Kaplan–Meier survival curves for the two groups. After PSM, the high UCR (≥27.208) group still exhibited an obviously higher in-hospital mortality (Figure 2B) compared to the low UCR (<27.208) group (*p* = 0.004). 

Furthermore, the association of two groups remained stable after PSM analyses using SMRW, PA, OW and adjusted propensity score. The values of the ORs were in the range of 2.43–2.594 and all *p* < 0.05 (Table 5).

### 3.8. Subgroup Analysis

The subgroup analysis was conducted to determine how the UCR affected the in-hospital mortality from various perspectives including age (<65 and ≥65 years old), sex, myocardial infarction, congestive heart failure, peripheral vascular disease, chronic pulmonary disease, renal disease, malignant cancer, mild liver disease, diabetes, SOFA (<3 and ≥3), sepsis, and WFNS grade (Figure 4). The high UCR (≥27.208) group presented a higher in-hospital mortality rate compared to the low UCR (<27.208) group in all subgroups. We analyzed the interactions between UCR and all subgroup factors and found no obvious interaction (*p* > 0.05).

## 4. Discussion

This study included 961 patients with non-traumatic subarachnoid hemorrhage whose information was extracted from the MIMIC-Ⅳ database. We performed univariable regression analysis, multivariable regression analysis, and PSM to reduce interference from possible confounding factors on in-hospital mortality. This large, retrospective cohort study suggested that, as a categorical or continuous variable, patients with high levels of UCR were more likely to have a higher risk of in-hospital mortality than patients with low levels of UCR. Furthermore, we found that the serum UCR level and in-hospital mortality had no interaction between subgroups. This is the first study to investigate the influence of the UCR on the prognosis of non-traumatic subarachnoid hemorrhage.

In the clinical environment, the UCR is a simple and commonly used index because it only requires venous blood, which is why the BUN is routinely measured in subarachnoid hemorrhage patients admitted to the ICU. Therefore, previous studies have explored the relationship between the serum BUN level and the worse prognosis of severe patients. Deng et al. conducted a study of 1738 acute ischemic stroke patients and found that higher UCR levels were related to a higher risk of poor three-month outcomes [20]. The study by Smita Mohanty et al. demonstrated that a level of UCR > 15 at admission was a significant independent predictor for neurological deterioration in ischemic stroke patients [21]. In addition, Zhu et al. conducted a study of 509 hospitalized patients with acute heart failure and determined that UCR was an independent predictor of all-cause mortality and that elevated UCR was related to poor prognosis [22]. Moreover, Han et al. used the MIMIC-III to determine the relationship between UCR and all-cause mortality in septic shock patients and found that a higher UCR was associated with increased mortality in these patients [8]. All the above results indicate that a higher UCR is associated with more serious conditions than a lower UCR. Our results are consistent with the above findings. This retrospective cohort study involved 961 patients, and the cut-off value of the UCR was considered to divide them into two groups. Compared with low UCR group, high UCR group exhibited higher in-hospital mortality (17.1 vs. 42.7%, respectively, *p* < 0.001). After adjustments for the confounding factors, our multivariable logistic regression analysis revealed that the in-hospital mortality for the high UCR (≥27.208) group remained higher than for the low UCR (<27.208) group both before and after PSM.

It is hard to identify the exact mechanisms behind the close correlation between serum UCR and in-hospital mortality in patients with subarachnoid hemorrhage. However, we can propose several hypothesized mechanisms to explain the relationship. Firstly, subarachnoid hemorrhage usually triggers a stress response in the body, which can lead to a disorder in the internal environment. Furthermore, unstable blood flow of the brain and kidney will lead to the change of BUN and creatinine. Secondly, previous studies suggested that the UCR was a routinely available indicator of hydration [10,23]. Dehydration is a very common phenomenon in ischemic stroke and hemorrhagic stroke, which is related to a high risk of poor outcomes at hospital discharge [24]. In the early stage of subarachnoid hemorrhage, consciousness disorder or dysphagia are the main causes of dehydration, which may lead to aggravation of the disease. 

Our research has the following advantages: (1) a large sample size and improved statistical reliability and (2) the missing value of UCR was low, which may reduce the selection bias. In addition, the findings of our study may help clinicians identify high-risk patients with non-traumatic subarachnoid hemorrhage. Despite the value of our findings, there were still some limitations. First, this was a single-center study, and multicenter studies are necessary to verify the accuracy of our conclusions. Second, the data on the UCR were collected during the first 24 h of the patient’s admission to the ICU and the dynamic changes of UCR could not be analyzed. Third, the optimal cut-off value of the UCR was calculated using ROC curve analysis, with the maximum Youden index used to predict the survival status. The UCR was considered applicable to divide patients into a low UCR (<27.208) group and a high UCR (≥27.208) group. However, the AUC was 0.564 (95% CI, 0.515–0.613), which was lower than expected. Therefore, it is necessary to verify the results through further studies.

## 5. Conclusions

In conclusion, this was the first study to investigate the prognostic significance of the UCR in non-traumatic subarachnoid hemorrhage patients. A high UCR was associated with a higher risk of in-hospital mortality than a low UCR. Therefore, the UCR can serve as a prognostic predictor of clinical outcomes in non-traumatic subarachnoid hemorrhage patients.

## Figures and Tables

**Figure 1 jcm-11-07031-f001:**
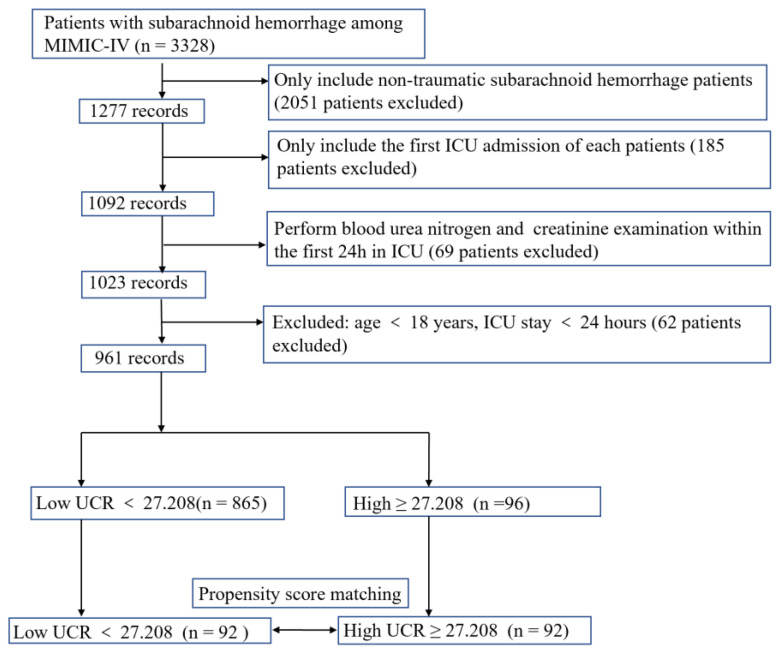
The flow chart of the study. MIMIV-IV, Medical Information Mart for Intensive Care IV; ICU, intensive care unit; UCR, urea nitrogen to creatinine ratio.

**Figure 2 jcm-11-07031-f002:**
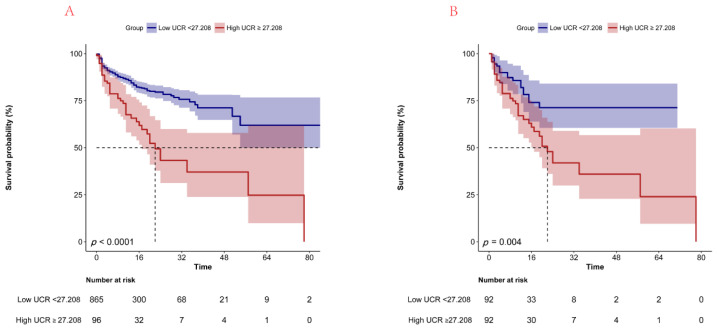
Kaplan–Meier survival curves of in-hospital mortality classified into two groups according to UCR before (**A**) and after PSM (**B**). UCR, urea nitrogen to creatinine ratio.

**Figure 3 jcm-11-07031-f003:**
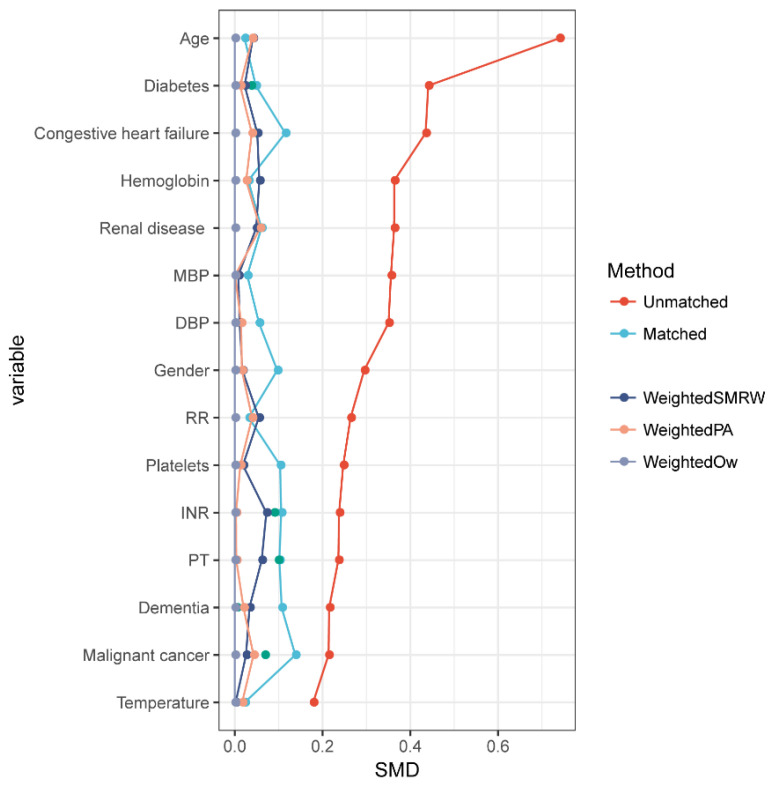
The results of matching. A standardized mean difference (SMD) was used to examine the degree of PSM. A threshold of less than 0.1 was considered acceptable. PSM, propensity score matching [19]; weighted SMRW, weighted the standardized mortality ratio weighting [16]; weighted PA, weighted pairwise algorithmic [15]; weighted OW, weighted overlap weight [18]. MBP, mean blood pressure; DBP, diastolic blood pressure; RR, respiratory rate; INR, international normalized ratio; PT, prothrombin time.

**Figure 4 jcm-11-07031-f004:**
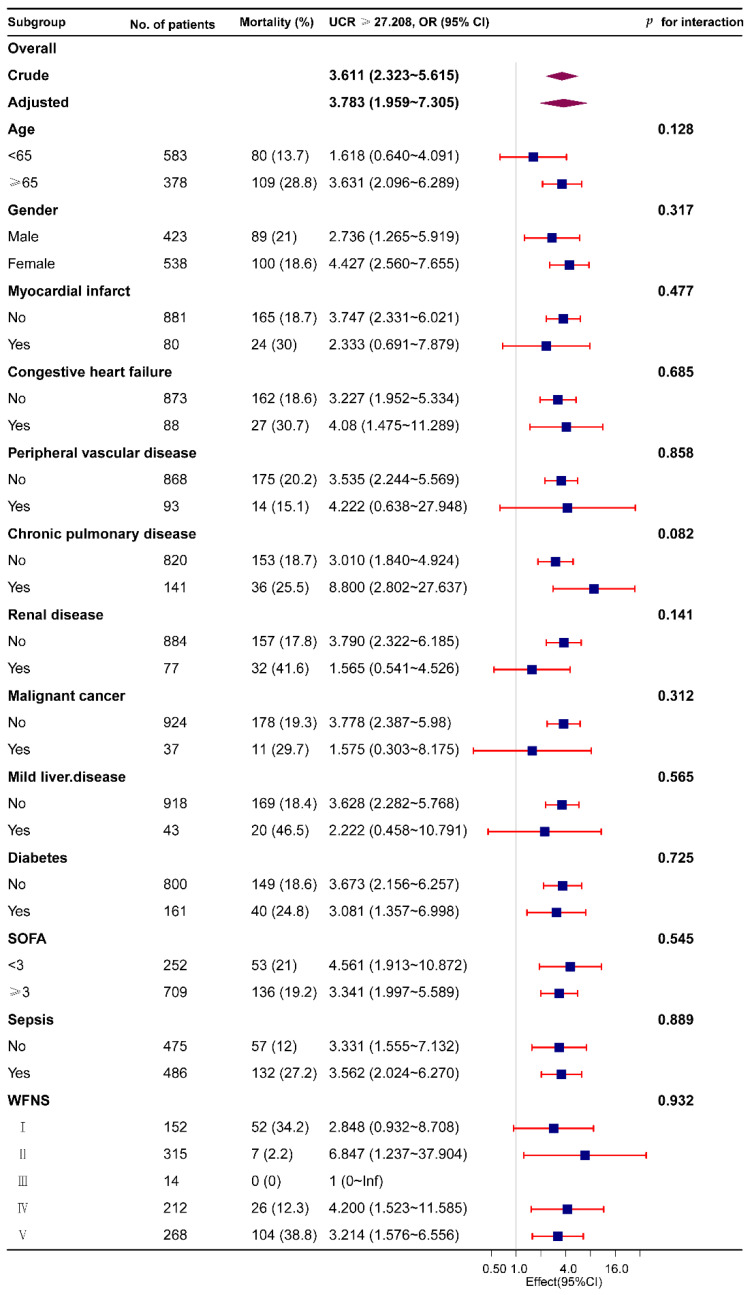
The relationship between the UCR and in-hospital mortality in subgroup analysis. CI, confidence interval; UCR, urea nitrogen to creatinine ratio; WFNS, World Federation of Neurosurgical Societies.

**Table 1 jcm-11-07031-t001:** The baseline clinical characteristics of patients with non-traumatic subarachnoid hemorrhage.

Variables	Total (n = 961)	Survival (n = 772)	Non-Survival (n = 189)	*p*-Value
Demographic				
Female, n (%)	538 (56.0)	438 (56.7)	100 (52.9)	0.342
Age, years	60.0 (51.0, 72.0)	58.5 (50.0, 70.0)	68.0 (56.0, 79.0)	<0.001
Ethnicity, n (%)				<0.001
Asian	36 (3.7)	23 (3)	13 (6.9)	
White	580 (60.4)	499 (64.6)	81 (42.9)	
Other	345 (35.9)	250 (32.4)	95 (50.3)	
Vital signs				
HR, beats/minute	78.0 (71.0, 87.0)	77.0 (70.0, 86.0)	82.0 (75.0, 92.0)	<0.001
SBP, mmHg	125.0 (115.0, 133.0)	125.0 (115.0, 133.0)	126.0 (116.0, 134.0)	0.741
DBP, mmHg	64.0 (58.0, 69.0)	64.0 (58.0, 69.0)	62.0 (57.0, 70.0)	0.136
MBP, mmHg	82.0 (76.0, 88.0)	82.0 (76.0, 88.0)	81.0 (76.0, 88.0)	0.582
RR, times/minute	18.0 (16.0, 20.0)	17.0 (16.0, 19.0)	19.0 (17.0, 21.0)	<0.001
Temperature, °C	37.0 (36.8, 37.3)	37.0 (36.8, 37.2)	37.0 (36.6, 37.5)	0.842
SpO2, %	98.0 (96.0, 99.0)	98.0 (96.0, 99.0)	98.0 (97.0, 99.0)	0.001
Comorbidities, n (%)				
Myocardial infarction	80 (8.3)	56 (7.3)	24 (12.7)	0.015
Congestive heart failure	88 (9.2)	61 (7.9)	27 (14.3)	0.006
Peripheral vascular disease	93 (9.7)	79 (10.2)	14 (7.4)	0.239
Chronic pulmonary disease	141 (14.7)	105 (13.6)	36 (19)	0.058
Peptic ulcer disease	6 (0.6)	4 (0.5)	2 (1.1)	0.336
Paraplegia	156 (16.2)	124 (16.1)	32 (16.9)	0.772
Renal disease	77 (8.0)	45 (5.8)	32 (16.9)	<0.001
Malignant cancer	37 (3.9)	26 (3.4)	11 (5.8)	0.116
Mild liver disease	43 (4.5)	23 (3)	20 (10.6)	<0.001
Diabetes	161 (16.8)	121 (15.7)	40 (21.2)	0.07
Vasospasm	80 (8.3)	75 (9.7)	5 (2.6)	0.002
DCI	66 (6.9)	60 (7.8)	6 (3.2)	0.025
Sepsis	486 (50.6)	354 (45.9)	132 (69.8)	<0.001
Laboratory results				
Urea nitrogen, mg/dL	13.0 (10.0, 18.0)	13.0 (10.0, 17.0)	18.0 (13.0, 28.0)	<0.001
Creatinine, mg/dL	0.8 (0.6, 1.0)	0.8 (0.6, 0.9)	1.0 (0.7, 1.3)	<0.001
UCR	17.0 (13.3, 21.8)	16.7 (13.3, 21.4)	18.0 (13.8, 25.0)	0.006
Hemoglobin, g/L	12.9 (11.6, 14.1)	13.0 (11.8, 14.2)	12.5 (10.9, 14.1)	0.032
Platelets, 10^9^/L	227.0 (184.0, 280.0)	230.0 (189.0, 281.0)	219.0 (152.0, 272.0)	<0.001
WBC, 10^9^/L	12.9 (9.7, 16.6)	12.5 (9.4, 15.9)	15.2 (11.6, 19.7)	<0.001
Anion gap, mmol/L	16.0 (14.0, 18.0)	16.0 (14.0, 18.0)	18.0 (15.0, 20.0)	<0.001
Bicarbonate, mmol/L	24.0 (22.0, 26.0)	24.0 (22.0, 26.0)	23.0 (21.0, 26.0)	0.039
Calcium, mg/dL	8.7 (8.3, 9.2)	8.7 (8.4, 9.1)	8.7 (8.2, 9.2)	0.855
Chloride, mmol/L	107.0 (104.0, 110.0)	107.0 (104.0, 109.0)	109.0 (105.0, 116.0)	<0.001
Sodium, mmol/L	141.0 (139.0, 144.0)	141.0 (139.0, 143.0)	144.0 (140.0, 149.0)	<0.001
INR	1.1 (1.1, 1.2)	1.1 (1.1, 1.2)	1.2 (1.1, 1.4)	<0.001
PT, s	12.6 (11.8, 13.8)	12.5 (11.7, 13.5)	13.6 (12.3, 15.8)	<0.001
APTT, s	29.0 (26.1, 33.5)	28.7 (25.9, 32.8)	30.2 (26.8, 37.2)	0.003
ALT, IU/L	78.0 (32.0, 78.0)	78.0 (42.0, 78.0)	78.0 (25.0, 78.0)	0.007
AST, U/L	127.0 (45.0, 127.0)	127.0 (52.0, 127.0)	127.0 (37.0, 127.0)	0.118
Glucose, mg/dL	131.5 (112.5, 156.3)	128.0 (109.8, 150.0)	151.6 (126.0, 187.8)	<0.001
Scores				
OASIS	31.0 (25.0, 40.0)	29.0 (23.0, 37.0)	41.0 (35.0, 46.0)	<0.001
GCS	13.0 (7.0, 14.0)	13.0 (8.0, 14.0)	7.0 (3.0, 15.0)	<0.001
SOFA	3.0 (2.0, 3.0)	3.0 (2.0, 3.0)	3.0 (2.0, 4.0)	0.001
WFNS Grade, n (%)				<0.001
I	152 (15.8)	100 (13)	52 (27.5)	
II	315 (32.8)	308 (39.9)	7 (3.7)	
II	14 (1.5)	14 (1.8)	0 (0)	
IV	212 (22.1)	186 (24.1)	26 (13.8)	
V	268 (27.9)	164 (21.2)	104 (55)	
Outcomes				
Length of ICU stay, days	12.0 (7.0, 20.0)	13.0 (8.0, 21.0)	5.0 (2.0, 13.0)	<0.001
Length of hospital stay, days	7.0 (3.0, 13.0)	7.0 (3.0, 13.0)	4.0 (2.0, 10.0)	<0.001

**Table 2 jcm-11-07031-t002:** The clinical characteristics of non-traumatic subarachnoid hemorrhage patients before PSM.

Characteristic	Before PSM			
	All Patients	Low UCR < 27.208	High UCR ≥ 27.208	*p*
N	961	865	96	
Demographic				
Female, n (%)	538 (56.0)	472 (54.6)	66 (68.8)	0.008
Age, years	60.0 (51.0, 72.0)	59.0 (50.0, 70.0)	74.0 (61.0, 80.0)	<0.001
Ethnicity, n (%)				0.161
Asian	36 (3.7)	29 (3.4)	7 (7.3)	
White	580 (60.4)	522 (60.3)	58 (60.4)	
Other	345 (35.9)	314 (36.3)	31 (32.3)	
Vital signs				
HR, beats/minute	78.0 (71.0, 87.0)	78.0 (70.0, 87.0)	81.5 (71.0, 92.0)	0.047
SBP, mmHg	125.0 (115.0, 133.0)	125.0 (115.0, 133.0)	124.5 (116.0, 132.2)	0.523
DBP, mmHg	64.0 (58.0, 69.0)	64.0 (58.0, 70.0)	60.0 (55.8, 66.0)	<0.001
MBP, mmHg	82.0 (76.0, 88.0)	82.0 (76.0, 88.0)	78.0 (75.0, 83.0)	<0.001
RR, times/minute	18.0 (16.0, 20.0)	18.0 (16.0, 20.0)	18.5 (17.0, 20.0)	0.015
Temperature, °C	37.0 (36.8, 37.3)	37.0 (36.8, 37.3)	36.9 (36.7, 37.1)	0.013
SpO2, %	98.0 (96.0, 99.0)	98.0 (96.0, 99.0)	98.0 (96.0, 99.0)	0.571
Comorbidities, n (%)				
Myocardial infarction	80 (8.3)	67 (7.7)	13 (13.5)	0.051
Congestive heart failure	88 (9.2)	66 (7.6)	22 (22.9)	<0.001
Peripheral vascular disease	93 (9.7)	88 (10.2)	5 (5.2)	0.119
Dementia	17 (1.8)	12 (1.4)	5 (5.2)	0.021
Paraplegia	156 (16.2)	147 (17)	9 (9.4)	0.055
Renal disease	77 (8.0)	59 (6.8)	18 (18.8)	<0.001
Malignant cancer	37 (3.9)	29 (3.4)	8 (8.3)	0.025
Mild liver disease	43 (4.5)	35 (4)	8 (8.3)	0.066
Diabetes	161 (16.8)	129 (14.9)	32 (33.3)	<0.001
DCI	66 (6.9)	64 (7.4)	2 (2.1)	0.051
Sepsis	486 (50.6)	429 (49.6)	57 (59.4)	0.069
Vasospasm, n (%)	80 (8.3)	77 (8.9)	3 (3.1)	0.052
Laboratory results				
Hemoglobin, g/L	12.9 (11.6, 14.1)	13.0 (11.7, 14.2)	12.2 (10.3, 13.6)	<0.001
Platelets, 10^9^/L	227.0 (184.0, 280.0)	229.0 (187.0, 283.0)	209.5 (154.8, 251.2)	<0.001
WBC, 10^9^/L	12.9 (9.7, 16.6)	12.9 (9.6, 16.6)	12.4 (9.9, 16.9)	0.658
Anion gap, mmol/L	16.0 (14.0, 18.0)	16.0 (14.0, 18.0)	16.0 (14.8, 19.0)	0.095
Bicarbonate, mmol/L	24.0 (22.0, 26.0)	24.0 (22.0, 26.0)	25.0 (23.0, 26.0)	0.083
Calcium, mg/dL	8.7 (8.3, 9.2)	8.7 (8.3, 9.2)	8.8 (8.3, 9.3)	0.478
Chloride, mmol/L	107.0 (104.0, 110.0)	107.0 (104.0, 110.0)	107.0 (103.0, 112.0)	0.919
Sodium, mmol/L	141.0 (139.0, 144.0)	141.0 (139.0, 143.0)	141.0 (138.0, 146.0)	0.520
INR	1.1 (1.1, 1.2)	1.1 (1.1, 1.2)	1.2 (1.1, 1.4)	<0.001
PT, s	12.6 (11.8, 13.8)	12.6 (11.7, 13.7)	13.2 (12.0, 15.3)	0.003
APTT, s	29.0 (26.1, 33.5)	28.9 (26.0, 33.2)	29.5 (26.6, 34.7)	0.406
ALT, IU/L	78.0 (32.0, 78.0)	78.0 (32.0, 78.0)	78.0 (31.8, 78.0)	0.686
AST, U/L	127.0 (45.0, 127.0)	127.0 (45.0, 127.0)	127.0 (43.8, 127.0)	0.443
Glucose, mg/dL	131.5 (112.5, 156.3)	131.3 (112.0, 154.3)	135.1 (116.5, 169.6)	0.097
Scores				
SOFA	3.0 (2.0, 3.0)	3.0 (2.0, 3.0)	3.0 (2.8, 3.0)	0.162
GCS	13.0 (7.0, 14.0)	13.0 (7.0, 14.0)	10.0 (6.0, 13.0)	0.006
OASIS	31.0 (25.0, 40.0)	31.0 (24.0, 39.0)	37.5 (29.8, 44.0)	<0.001
WFNS, n (%)				0.01
I	152 (15.8)	138 (16)	14 (14.6)	
II	315 (32.8)	296 (34.2)	19 (19.8)	
III	14 (1.5)	11 (1.3)	3 (3.1)	
IV	212 (22.1)	190 (22)	22 (22.9)	
V	268 (27.9)	230 (26.6)	38 (39.6)	
Outcomes				
In-hospital mortality, n (%)	189 (19.7)	148 (17.1)	41 (42.7)	<0.001
Length of ICU stay, days	12.0 (7.0, 20.0)	12.0 (7.0, 20.0)	11.0 (5.0, 19.0)	0.195
Length of hospital stay, days	7.0 (3.0, 13.0)	7.0 (3.0, 13.0)	5.5 (2.0, 11.2)	0.093

**Table 3 jcm-11-07031-t003:** Multivariable logistic regression analyses for in-hospital mortality in patients with non-traumatic subarachnoid hemorrhage.

Characteristic	Crude Model		Model I		Model II		Model III	
	OR (95% CI)	*p*-Value	OR (95% CI)	*p*-Value	OR (95% CI)	*p*-Value	OR (95% CI)	*p*-Value
Before PSM								
UCR	1.038 (1.018~1.059)	<0.001	1.030 (1.008~1.052)	0.0071	1.031 (1.009~1.054)	0.0062	1.038 (1.009~1.068)	0.0102
Low UCR (<27.208)	1(Ref)		1(Ref)		1(Ref)		1(Ref)	
High UCR (≥27.208)	3.611 (2.323~5.615)	<0.001	3.110 (1.937~4.995)	<0.001	2.979 (1.818~4.844)	<0.001	3.783 (1.959~7.305)	<0.001
After PSM								
UCR	1.039 (1.008~1.071)	0.0014	1.036 (1.003~1.070)	0.0326	1.041 (1.006~1.078)	0.0215	1.066 (1.011~1.124)	0.0189
Low UCR (<27.208)	1 (Ref)		1 (Ref)		1 (Ref)		1 (Ref)	
High UCR (≥27.208)	2.995 (1.540~5.671)	0.0011	3.082 (1.515~6.271)	0.0019	3.634 (1.673~7.892)	0.0011	10.161 (2.691~38.368)	<0.001

OR, odds ratio; CI, confidence interval; PSM, propensity score matching; UCR, urea nitrogen to creatinine ratio.

**Table 4 jcm-11-07031-t004:** The clinical characteristics of non-traumatic subarachnoid hemorrhage after PSM.

Characteristic	After PSM			
	All Patients	Low UCR < 27.208	High UCR ≥ 27.208	*p*
N	184	92	92	
Demographic				
Female, n (%)	132 (71.7)	68 (73.9)	64 (69.6)	0.513
Age, years	72.0 (60.0, 81.0)	72.0 (59.8, 81.0)	72.5 (60.8, 80.0)	0.781
Ethnicity, n (%)				0.092
Asian	8 (4.3)	1 (1.1)	7 (7.6)	
White	113 (61.4)	57 (62)	56 (60.9)	
Other	63 (34.2)	34 (37)	29 (31.5)	
Vital signs				
HR, beats/minute	79.5 (71.0, 90.0)	78.5 (71.8, 88.2)	81.5 (71.0, 92.2)	0.326
SBP, mmHg	124.0 (116.0, 132.0)	124.0 (115.8, 129.0)	125.5 (116.0, 133.2)	0.489
DBP, mmHg	61.0 (56.0, 66.2)	62.0 (56.8, 67.0)	60.0 (55.8, 66.0)	0.439
MBP, mmHg	78.0 (74.0, 84.0)	79.0 (74.0, 86.0)	78.0 (74.8, 83.0)	0.878
RR, times/minute	18.0 (17.0, 21.0)	18.0 (17.0, 21.0)	19.0 (17.0, 20.0)	0.838
Temperature, °C	37.0 (36.7, 37.3)	37.0 (36.8, 37.3)	36.9 (36.7, 37.1)	0.152
SpO2, %	97.0 (96.0, 99.0)	97.0 (96.0, 99.0)	97.0 (96.0, 99.0)	0.684
Comorbidities, n (%)				
Myocardial infarction	21 (11.4)	11 (12)	10 (10.9)	0.817
Congestive heart failure	32 (17.4)	14 (15.2)	18 (19.6)	0.437
Peripheral vascular disease	12 (6.5)	7 (7.6)	5 (5.4)	0.55
Cerebrovascular disease	184 (100.0)	92 (100)	92 (100)	1
Dementia	8 (4.3)	5 (5.4)	3 (3.3)	0.72
Paraplegia	31 (16.8)	22 (23.9)	9 (9.8)	0.01
Renal disease	28 (15.2)	13 (14.1)	15 (16.3)	0.681
Malignant cancer	11 (6.0)	4 (4.3)	7 (7.6)	0.351
Mild liver disease	12 (6.5)	4 (4.3)	8 (8.7)	0.232
Diabetes	54 (29.3)	26 (28.3)	28 (30.4)	0.746
Hypertension	4 (1.1)	2 (1.1)	2 (1.1)	1
DCI	7 (3.8)	5 (5.4)	2 (2.2)	0.444
Sepsis	103 (56.0)	50 (54.3)	53 (57.6)	0.656
Vasospasm, n (%)	6 (3.3)	3 (3.3)	3 (3.3)	1
Laboratory results				
Hemoglobin, g/L	12.4 (10.9, 13.6)	12.6 (11.4, 13.3)	12.3 (10.7, 13.7)	0.680
Platelets, 10^9^/L	210.0 (159.0, 255.2)	209.0 (162.5, 265.0)	210.0 (156.8, 251.2)	0.468
WBC, 10^9^/L	13.2 (10.2, 17.1)	13.3 (10.5, 16.9)	12.8 (10.0, 17.2)	0.986
Anion gap, mmol/L	16.0 (14.0, 18.2)	16.5 (14.0, 18.0)	16.0 (14.0, 19.0)	0.605
Bicarbonate, mmol/L	25.0 (22.0, 26.0)	25.0 (22.0, 27.0)	25.0 (23.0, 26.0)	0.293
Calcium, mg/dL	8.8 (8.3, 9.3)	8.8 (8.3, 9.3)	8.8 (8.3, 9.3)	0.964
Chloride, mmol/L	107.0 (103.0, 111.0)	108.0 (104.0, 109.2)	107.0 (103.0, 112.2)	0.970
Sodium, mmol/L	141.0 (139.0, 144.0)	141.0 (139.0, 144.0)	141.0 (138.0, 146.0)	0.826
INR	1.2 (1.1, 1.3)	1.2 (1.1, 1.3)	1.2 (1.1, 1.3)	0.990
PT, s	13.2 (11.9, 14.9)	13.3 (11.9, 14.8)	13.2 (12.0, 14.9)	1.000
APTT, s	29.2 (26.5, 34.1)	29.1 (26.4, 34.0)	29.4 (26.6, 34.4)	0.847
ALT, IU/L	78.0 (25.0, 78.0)	78.0 (22.2, 78.0)	78.0 (31.0, 78.0)	0.417
AST, U/L	127.0 (36.8, 127.0)	127.0 (35.8, 127.0)	127.0 (42.8, 127.0)	0.146
Glucose, mg/dL	138.5 (117.3, 169.2)	141.6 (116.4, 166.9)	135.1 (118.6, 169.2)	0.833
Scores				
SOFA	3.0 (2.8, 3.0)	3.0 (2.8, 3.0)	3.0 (2.8, 3.0)	0.863
GCS	10.0 (7.0, 14.0)	10.0 (7.0, 14.0)	9.0 (6.0, 13.0)	0.161
OASIS	37.0 (29.0, 44.0)	36.5 (29.0, 44.0)	37.0 (29.0, 43.5)	0.600
WFNS, n (%)				0.142
I	27 (14.7)	14 (15.2)	13 (14.1)	
II	40 (21.7)	24 (26.1)	16 (17.4)	
III	3 (1.6)	0 (0)	3 (3.3)	
IV	49 (26.6)	27 (29.3)	22 (23.9)	
V	65 (35.3)	27 (29.3)	38 (41.3)	
Outcomes				
In-hospital mortality, n (%)	59 (32.1)	19 (20.7)	40 (43.5)	< 0.001
Length of ICU stay, days	6.0 (3.0, 13.0)	6.0 (3.0, 13.2)	6.0 (2.0, 12.0)	0.321
Length of hospital stay, days	11.0 (5.0, 19.2)	11.5 (6.0, 21.0)	10.5 (5.0, 18.2)	0.449

**Table 5 jcm-11-07031-t005:** Associations between UCR and in-hospital mortality in the crude analysis, multivariable analysis, and propensity-score analyses.

Analysis	In-Hospital Mortality	*p* Value
No. of results/no. of patients at risk (%)		<0.001
Low UCR (<27.208)	148/865 (17.1)	
High UCR (≥27.208)	41/96 (42.7)	
Crude analysis-odds ratio (95% CI)	3.611 (2.323~5.615)	<0.001
Multivariable analysis-odds ratio (95% CI)	2.663 (1.627~4.359)	<0.001
Adjusted propensity score	2.594 (1.615~4.164)	<0.001
With SMRW	2.536 (1.642~3.916)	<0.001
With PA	2.502 (1.335~4.689)	0.0042
With OW	2.431 (1.203~4.912)	0.0133

PA, pairwise algorithmic; SMRW, standardized mortality ratio weight; OW, overlap weight. CI, confidence interval; UCR, urea nitrogen to creatinine ratio.

## Data Availability

All data in the article can be obtained from the MIMIC-IV database (https://mimic.physionet.org/) (accessed on 17 October 2021).

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
