# Peer review of "Relationship between the Blood Urea Nitrogen to Creatinine Ratio and In-Hospital Mortality in Non-Traumatic Subarachnoid Hemorrhage Patients: Based on Propensity Score Matching Method"

_jcm, 2022, doi:10.3390/jcm11237031_

Round 1

Reviewer 1 Report

Title: ‘Relationship Between the Blood Urea Nitrogen to Creatinine 2 Ratio and in-hospital Mortality in Subarachnoid Hemorrhage 3 Patients: Based on Propensity Score Matching Method’.

The authors aimed to explore the correlation between the blood urea nitrogen to creatinine ratio (UCR) and in-hospital mortality in subarachnoid hemorrhage patients. For that purpose they analyzed a total of 1598 patients from a retrospective cohort study. The analysis was performed in the whole population and in a propensity score-matching (PSM) cohort. They found that high UCR levels were associated with increased mortality in unadjusted and adjusted models both in the whole population and in the PSM cohort

UCR is an easily accessible biomarker in the neurocritical population and it use as a prognostic biomarker might be of interest from a practical point of view. I have several concerns to be addressed by the authors.

The English language should be thoroughly edited. Some parts of the manuscript are difficult to follow (e.g. the description of the statistical methods). I would recommend the authors to improve the readability of their manuscript.

This study is focused in subarachnoid hemorrhage (SAH) patients. Data on the etiology of SAH (traumatic vs spontaneous) or in the presence or absence of intracerebral aneurysms in the case of spontaneous SAH is lacking and this is a critical limitation. The authors should clarify this data. Moreover, if the study is focused in spontaneous SAH, specific data on treatment-related methods (coiling, clipping, etcetera) or in specific in-hospital complications (such as rebleeding, vasospasm, hydrocephalus, delayed cerebral ischemia, etcetera) should also be described (along with the association of UCR with this specific complications). The lack of this information is also a major limitation of the study.

The description of the statistical methods, including that referred to the multivariate analyses is difficult to follow. The authors should improve and synthetize that description. Moreover, the matching methods explored in this cohort are illustrated in figure 4 (PSM, IPTW and so on) but this is not explained in the methods section where the authors only describe the PSM approach.

The results section is relatively large and the information is somewhat redundant to that presented in the tables. I would recommend the authors to summarize their findings. The authors should also review the Table 3 (multivariate analyses). The OR of the dichotomized UCR seems to be interchanged between the high vs low category. If the OR is higher than 1 in the continuous variable this would be the same for the high UCR category and not the opposite. In Figure 3, I would suggest adding also a subgroup based in admission WFNS.

In the discussion section, the authors should discuss with more details how they interpret the low AUC of UCR obtained in the ROC analysis. More data on the accuracy of this biomarker as predictive values and overall accuracy need to be clearly exposed in the results. A biomarker with such a low sensibility could be a good biomarker? Please, extend the discussion on this issue.

The plausible mechanisms explaining the association between UCR and mortality are very general. The last sentence of that paragraph (lines 284-285) where the authors state ‘Therefore, the increase of cerebral blood flow based on UCR value may help to improve the prognosis of subarachnoid hemorrhage’ should be deleted because the data presented herein does not support that statement. The observation extracted from this study is only an association and mechanistic or causal inferences could not be extracted. Eventually, the limitations of the study should include the poor definition of the SAH population, although without more specific data on SAH subtypes or specific complications the external validity of this data is critically compromised.

Reviewer 2 Report

Wang et. al present an interesting manuscript about the impact of blood urea nitrogen to creatine ratio (UCR) in SAH patients. The authors conclude that a UCR ≥ 27.208 is associated with a increased in-hospital mortality in these patients. In accordance to previous studies about serum biomarkers (e.g. fibrinogen to albumin ratio or CRP to albumin ratio), the UCR seems to be an additional serum biomarker to predict intra-hospital mortality in SAH patients.

The manuscript is well written, however there are several issues, especially in the study design:

1) Inclusion criteria:  Why did the authors not exclude patients with chronic renal and liver diseases/dysfunction?  Both comorbidities had an essential impact on the UCR. In addition, what's about patients with traumatic subarachnoid hemorrhage and non-aneurysmal subarachnoid hemorrhage? Both forms of subarachnoid hemorrhage had a better outcome, than patients with aneurysmal subarachnoid hemorrhage. Where patients with traumatic and non-aneurysmal subarachnoid hemorrhage excluded? It is not senseful to mix patients with aneurysmal, non-aneurysmal and traumatic subarachnoid hemorrhage based on their different pathophysiology, outcome and mortality.

2) I am surprised that the occurrence of DCI and vasospasm hat no influence on the mortality rate of the SAH patients. This is in contrast to most of publications about mortality in SAH patients. The authors should explain these surprising results.

3) Why did the authors used the SOFA score, which is usually used for septic patients? I think the NIHSS score will be more suitable

4) Did the authors analyzed more serum biomarkers and ratios, which are predictive for mortality in SAH patients, e.g. troponin, cortisol, albumin or CRP/albumin ratio?

5) Where there any difference between survivor and non-survivor with respect to the treatment regime (clipping vs. coiling)? Where all aneurysm treated within the first 24 hours?

Round 2

Reviewer 2 Report

As suggested in the initial review: The mortality rate differs significantly between patients with traumatic subarachnoid hemorrhage, aneurysmal subarachnoid hemorrhage and subarachnoid hemorrhage without evident aneurysm.  In my mind, it is impossible to analyses all these patients, as one population, concerning intra-hospital mortality.

However, it is unfeasible to the authors to improve this major issue. They conclude: “ Therefore, it is necessary to verify the results through further studies. Lastly, this was a retrospective observational study unfortunately, the free and publicly available database did not specify which belongs to  the traumatic subarachnoid hemorrhage or spontaneous subarachnoid hemorrhage, more  studies were needed to validate the accuracy of the results”.

Before recommendation a publication, the manuscript has to improve in this elementary issue.
